# Adversarial Attacks and Defenses on Graph-aware Large Language Models

## Abstract

Large Language Models (LLMs) are increasingly integrated with graph-structured data for tasks like node classification, a domain traditionally dominated by Graph Neural Networks (GNNs). While this integration leverages rich relational information to improve task performance, its robustness against adversarial attacks remains unexplored. We take the first step to explore the vulnerabilities of graph-aware LLMs by leveraging existing adversarial attack methods tailored for graph-based models, including those for poisoning (training-time attacks) and evasion (test-time attacks), on two representative models, LLaGA and GraphPrompter. Additionally, we discover a new attack surface for LLaGA where an attacker can inject malicious nodes as placeholders into the node sequence template to severely degrade performance. Our systematic analysis reveals that certain design choices in graph encoding can enhance attack success, in particular: (1) the node sequence template in LLaGA increases its vulnerability; (2) the GNN encoder used in GraphPrompter demonstrates greater robustness; and (3) both approaches remain susceptible to imperceptible feature perturbation attacks. Finally, we propose an end-to-end defense framework GalGuard, that combines an LLM-based feature correction module to mitigate feature-level perturbations and GNN-based defenses to protect against structural attacks.

## 1 Introduction

Graphs, characterized by nodes and edges that represent entities and relationships, are used to model complex structures in various real-world domains, including social networks (Fan et al., 2019), biology (Dong et al., 2022), finance (Wang et al., 2021), and healthcare (Ahmedt-Aristizabal et al., 2021). Graph Neural Networks (GNNs) (Hamilton et al., 2017; Kipf & Welling, 2022; Veličković et al., 2018), designed specifically for graph-structured data, excel in tasks such as node classification and link prediction. Recently, the success of large language models (LLMs) such as GPT-4 (Achiam et al., 2023), Gemini (Team et al., 2023), LLaMA (Touvron et al., 2023; Dubey et al., 2024), Vicuna (Chiang et al., 2023), and PaLM-2 (Anil et al., 2023), trained on vast amounts of textual data, has sparked interest in their potential to enhance graph-related tasks. This fusion of graph data with LLMs (called *graph-aware LLMs* (Xie et al., 2023)) has opened a new line of research, where LLMs are increasingly being utilized to perform tasks traditionally handled by GNNs (Ye et al., 2023; Fatemi et al., 2023; Kuang et al., 2023; Zhang et al., 2023; Chen et al., 2023b; 2024). In fact, recent research has shown that graph-aware LLMs can outperform GNNs on various graph-related tasks (Wenkel et al., 2023; Chen et al., 2023b; Zhu et al., 2024; Chen et al., 2024), positioning them as powerful alternatives.

Despite their growing popularity, the vulnerabilities of graph-aware LLMs to adversarial attacks remain largely unexplored. Existing adversarial attacks on GNNs, including those for poisoning (training-time attacks) and evasion (test-time attacks) (Zügner et al., 2018; Zügner & Günnemann, 2019; Dai et al., 2023), operate under a black-box setting where the attacker has no knowledge of the model's internal workings. These attacks undermine *model integrity* by intentionally degrading model performance by subtly altering graph structure or node features. While GNN research has predominantly focused on structural attacks, the textual nature of graph-aware LLMs makes imperceptible perturbations to node features a significant concern that has been largely overlooked.

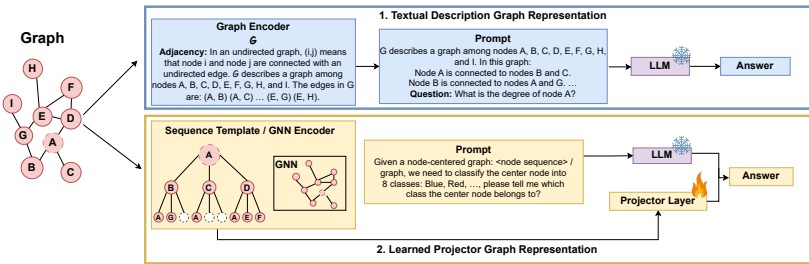

Figure 1: **Graph encoding-based adaptations of LLMs.** In the textual description graph representation, graphs are explained textually (top part). In the learned projector representation (bottom part), graphs are encoded either by node templates that turn the graph into sequences or by GNNs. Note that in both cases, the LLM is frozen. Only the projector that maps the graph into tokens is trained.

Motivated by three key observations: 1) the recent shift from GNNs to LLMs for graph tasks, 2) the black-box nature of current adversarial attacks on GNNs, and 3) the potentially new vulnerabilities to feature perturbations, we ask the critical question of whether *adversarial attacks designed for GNNs remain effective against the emerging graph-aware LLM paradigm*. In addition, the success of graph-aware LLMs in handling graph tasks *highly depends* on the effective integration of graphs into the LLM. While processing the node features is relatively straightforward given LLM's proficiency in handling textual data, incorporating the graph structure is more complex and requires innovative techniques to capture relational information (Chen et al., 2024; Guo et al., 2023; Liu & Wu, 2023; Zhang et al., 2023; Sun et al., 2023; Bi et al., 2024; Pan et al., 2024). Therefore, to effectively answer our research question, we first categorize existing graph-aware methods into two classes based on their approach to encoding graphs into LLMs; one that represents graph data as *textual descriptions*, converting graph information into natural language, and another that utilizes *learned projectors* to bridge the inherent differences between graph and text domains. While conventional text-based adversarial attacks on LLMs should generally suffice to compromise LLMs handling textual descriptions of graphs, we focus our investigation on learned projector methods to better understand the contribution of each modality, *i.e.,* graph or text, to attack success. Specifically, we investigate two computationally efficient representative models, LLAGA (Chen et al., 2024) and GRAPHPROMPTER (Liu et al., 2024), which both utilize frozen pre-trained LLM.

Our analysis reveals that certain design choices in graph encoding for LLMs can unexpectedly enhance the success of attacks. Notably, we identify a new attack surface in LLAGA that could allow an attacker to severely degrade the model's performance by injecting malicious nodes as placeholders within the node sequence template. In addition, we propose an end-to-end graph-aware LLM defense, GALGUARD, which integrates an LLM-based feature corrector with adapted GNN-based structural defenses to provide robust protection. We are the first to investigate the vulnerability of graph-aware LLMs to adversarial attacks. The contributions of the paper are summarized as follows:

- We create a taxonomy for graph-aware LLMs based on their encoding approaches to understand how different methodologies integrate graph structures into LLMs.
- We investigate the transferability of representative adversarial attacks on GNNs, *i.e.,* NET-TACK (Zügner et al., 2018) and METAATTACK (Zügner & Günnemann, 2019), to graph-aware LLMs.
- We discover a new attack surface in LLAGA, showing that injecting malicious nodes as placeholders into the node sequence template can degrade its performance.
- We provide a holistic assessment of the vulnerabilities of graph-aware LLMs by performing imperceptible attacks on textual node features, a perspective often overlooked attack in GNNs.
- We propose GALGUARD, a novel end-to-end defense strategy that integrates LLM-based feature correction with the adaptation of existing GNN-based structural defenses.

## 2 GRAPH ENCODING-BASED TAXONOMY OF LLM ADAPTATIONS

Graph encoding methods for LLMs can be categorized into two main approaches: *textual descriptions of graphs* (where graph information is encoded as natural language) and *learned graph projectors*

(where a projector is utilized to encode graph data into embeddings or structured forms that LLMs can directly process). We briefly describe the different encoding approaches in each category. Figure 5 provides the illustration of both approaches. A detailed description is in Appendix A.1 and the related work on GNN attacks is in Appendix A.2.

***Graph Representation as a Textual Description.*** This approach converts graph data into natural language, enabling LLMs to process them as textual descriptions, often with specific instructions to query LLMs (Guo et al., 2023; Liu & Wu, 2023; Ye et al., 2023; Wenkel et al., 2023; Wei et al., 2024b; Fatemi et al., 2023; Tan et al., 2023; Zhao et al., 2023; Wei et al., 2024a; Zhang, 2023). In multimodal applications, Liu et al. (2023) and Su et al. (2022) linked molecule graphs with textual descriptions. Furthermore, Huang et al. (2023a) investigated incorporating graph structure into LLM prompts. Their findings revelas that LLMs tend to process graph-related prompts as contextual paragraphs, underscoring the importance of prompt design, as seen in studies like (Brannon et al., 2023; Zhang et al., 2023; Sun et al., 2023; Bi et al., 2024; Pan et al., 2024; He et al., 2024). Other research has focused on using LLMs to enhance node features, graph structures, act as predictors, or serve as feature extractors for GNNs (Chen et al., 2023a; Chandra et al., 2020; Chen et al., 2023b; Mavromatis et al., 2023). For inference, Zhu et al. (2024) proposed a fine-tuning method for textual graph that combined LLMs and GNNs, and Duan et al. (2023) introduced an effective approach that enhances textual graph learning. While these methods offer a straightforward way to use LLMs for graph tasks, they often struggle to fully capture structural intricacies (especially for large graphs) and are heavily dependent on prompt engineering.

***Graph Representation as a Learned Projector.*** To address the limitations of textual representations, learned projector methods transform graph data into embeddings or graph vectors that LLMs can process more directly. This includes methods like LLAGA, which uses node-level templates and a linear projector (Chen et al., 2024), and others that incorporate GNN layers into a pre-trained LLM (Liu et al., 2024; Qin et al., 2023). Other approaches use graph neural prompts to encode knowledge graphs (Tian et al., 2024), develop prompt-based node feature extractors with a GNN adapter (Huang et al., 2023b), or condense graph information into fixed-length prefixes using graph transformers (Chai et al., 2023; Yang et al., 2021; Peng et al., 2024).

**This Work:** We focus on projector methods that utilize *frozen pre-trained LLM*s because they are computationally efficient and avoid the substantial costs of retraining (Chen et al., 2024; Liu et al., 2024; Qin et al., 2023; Tian et al., 2024; Huang et al., 2023b). For our investigation, we use two representative models: LLAGA (Chen et al., 2024), which employs node-level templates with a linear projector, and GRAPHPROMPTER (Liu et al., 2024), which uses a GNN with a linear projector. We do not consider textual description methods, as they often struggle with scalability and do not introduce the novel, graph-specific vulnerabilities we aim to investigate.

## 2.1 THREAT MODEL

We characterize our threat model based on established research on adversarial attacks in graphs, and extend its applicability to graph-aware LLMs (Zügner et al., 2018; Wu et al., 2019; Zügner & Günnemann, 2019; Ma et al., 2020; Jin et al., 2021; Dai et al., 2023; Li et al., 2023). For example, in social networks, attackers often control only a few bot accounts; this limited access aligns with our threat model, where adversaries manipulate a small subset of nodes to evade detection or influence legitimate user classification. Another example is Wikipedia, where hoax articles with sparse connections to legitimate ones can be strategically linked to manipulate the classification of genuine content, causing misclassification of real articles' categories. This transferability is particularly concerning as these advanced models have the potential of being deployed in critical applications such as recommendation system, social media analysis or even healthcare, potentially replacing GNNs. Thus, following existing works on adversarial attacks in graphs, we characterize the threat model along three dimensions; model access, adversarial capabilities, and attack strategies, followed by defining the attacker's goal.

**Model Access.** The attacker operates in a black-box setting, with no knowledge of the model's architecture or parameters. The attacker can only influence the model indirectly by interacting with the training or test data.

**Adversarial Capabilities.** Consistent with existing poisoning (evasion) attacks, the attacker has complete access to the training (inference) data of the target model. Within a predefined budget

Table 1: Performance of NETTACK and METAATTACK on Cora, Citeseer, and PubMed datasets for graph-aware LLMs LLaGA and GRAPHPROMPTER.

| Attack | Type | Cora | | Citeseer | | PubMed | |
|---|---|---|---|---|---|---|---|
| | | LLaGA | GRAPHPROMPTER | LLaGA | GRAPHPROMPTER | LLaGA | GRAPHPROMPTER |
| Clean | — | $0.89_{\pm 0.07}$ | $0.60_{\pm 0.03}$ | $0.71_{\pm 0.04}$ | $0.70_{\pm 0.03}$ | $0.90_{\pm 0.04}$ | $0.90_{\pm 0.02}$ |
| NETTACK | Poisoning | $0.87_{\pm 0.04}$ (2%) | $0.60_{\pm 0.03}$ (0%) | $0.64_{\pm 0.05}$ (10%) | $0.68_{\pm 0.02}$ (3%) | $0.89_{\pm 0.03}$ (1%) | $0.88_{\pm 0.02}$ (2%) |
| | Evasion | $0.55_{\pm 0.09}$ (38%) | $0.53_{\pm 0.05}$ (12%) | $0.59_{\pm 0.04}$ (19%) | $0.61_{\pm 0.03}$ (14%) | $0.84_{\pm 0.05}$ (7%) | $0.81_{\pm 0.02}$ (10%) |
| METAATTACK | Poisoning | $0.79_{\pm 0.03}$ (11%) | $0.58_{\pm 0.05}$ (3%) | $0.63_{\pm 0.04}$ (12%) | $0.65_{\pm 0.06}$ (7%) | $0.89_{\pm 0.08}$ (1%) | $0.85_{\pm 0.02}$ (6%) |
| | Evasion | $0.44_{\pm 0.06}$ (51%) | $0.52_{\pm 0.04}$ (13%) | $0.50_{\pm 0.03}$ (35%) | $0.56_{\pm 0.02}$ (22%) | $0.73_{\pm 0.04}$ (19%) | $0.77_{\pm 0.03}$ (14%) |

and under unnoticeability constraints (aiming to avoid detection), the attacker can modify or inject data. The attacker leverages two primary strategies: feature manipulation (altering node features) and structure manipulation (adding or removing edges).

**Attacker's Goal.** The attacker's goal is to compromise model integrity by degrading the overall performance, specifically by increasing the misclassification rate of the classification model.

In the following sections, we present our evaluation setup and then assess the success of adversarial attacks on graph-aware LLMs.

## 3 EXPERIMENTAL SETUP

**Dataset and Tasks.** We utilize three widely used graph datasets: Cora, Citeseer, and PubMed (Yang et al., 2016). These text-attributed graph datasets, which vary in size and sparsity from small to medium scales, are common benchmarks for models like those in our study (Chen et al., 2023a; 2024; Liu et al., 2024). To further validate scalability, we also include the larger ArXiv dataset (Hu et al., 2020) in an additional experiment (see Appendix A.6). We adopt standard train/validation/test splits of 6:2:2 for the three smaller datasets and 5:2:3 for ArXiv, as in (Chen et al., 2024). Detailed statistics are provided in Table 5. We use *node classification* as our primary task, a widely used method for evaluating graph machine learning models and the primary setting for adversarial attacks (Zügner et al., 2018; Zügner & Günnemann, 2019; Liu & Wu, 2023).

**Evaluation Metrics.** We employ *accuracy* as the evaluation metric, , which measures the proportion of correctly predicted labels. Each experiment is repeated three times, and we report the mean and standard deviation. All experiments were conducted on a machine equipped with NVIDIA A40 GPU with 48GB of memory.

**Graph-Aware LLM Methods.** We employ two representative graph-aware LLM methods: LLaGA (Chen et al., 2024) and GRAPHPROMPTER (Liu et al., 2024). Both methods utilize a *frozen pre-trained LLM*, making them computationally efficient and adaptable to various graph tasks and LLM architectures. LLaGA uses node-level templates with a linear projector, while GRAPH-PROMPTER employs a GNN-based encoder and a linear projector. In both cases, the projector is a 2-layer multi-layer perceptron (MLP). The foundational LLMs for LLaGA and GRAPHPROMPTER are Vicuna-7B (Chiang et al., 2023) and LLAMA2-7B (Touvron et al., 2023), respectively.

**Adversarial Attack Methods.** We employ the poisoning and evasion attack from two representative adversarial attack methods NETTACK (Zügner et al., 2018) and METAATTACK (Zügner & Günnemann, 2019) for conducting our investigation. NETTACK is a targeted attack that finds minimal perturbations to misclassify specific nodes, while METAATTACK performs untargeted attacks by treating the graph structure as a hyperparameter to optimize. For NETTACK, we adopt the approach of (Li et al., 2023; Zügner & Günnemann, 2019) by sequentially targeting each node and aggregating the results, effectively transforming it into an untargeted evaluation. For all experiments, we perturb only 10% of the edges to maintain the unnoticeability constraint, as in (Zügner et al., 2018; Zügner & Günnemann, 2019). The full details is in Appendix A.3.

## 4 ASSESSING THE VULNERABILITY OF GRAPH-AWARE LLMs TO ADVERSARIAL ATTACKS

In this section, we first assess the vulnerability of graph-aware LLMs to existing adversarial attacks on GNNs, including poisoning and evasion attacks (Section 4.1). Then, we analyze why LLAGA is more susceptible to graph adversarial attacks (Section 4.2). Furthermore, we propose several new adversarial attacks specifically tailored for graph-aware LLMs, with higher attack accuracies than NETTACK and METAATTACK (Sections 4.3 to 4.5). Finally, in Section 4.2, we study the impact of varying feature perturbation and the choice of LLM.

### 4.1 ADAPTING POISONING AND EVASION ATTACKS ON GRAPH-AWARE LLMs

We analyze poisoning and evasion attacks by adapting methods from NETTACK (Zügner & Günnemann, 2019) and METAATTACK (Zügner & Günnemann, 2019). For poisoning attacks on LLAGA, we use a perturbed graph to generate the node-level sequences and then retrain the projector. For poisoning attacks on GRAPHPROMPTER, we use the perturbed graph to re-encode the graph using a GAT and retrain its projector. For evasion attacks, both models are trained on unperturbed data, and perturbations are introduced only during the inference phase.

**Results.** Our experiments reveal several key insights, as summarized in Table 1. First, we observe that poisoning attacks have a limited impact on graph-aware LLMs, with a maximum performance degradation of 11% on both attack methods. This suggests that the models' robust representations and reliance on additional context from the LLM provide resilience. In contrast, evasion attacks prove highly effective, causing significant performance degradation of up to 51%. Secondly, for the different adversarial attack methods, METAATTACK generally outperforms NETTACK for both poisoning and evasion attacks across all datasets. This is attributed to METAATTACK's ability to create small but highly effective perturbations by treating graph edges as hyperparameters. Finally, we observe that denser datasets are more vulnerable to attacks than sparse ones. For instance, LLAGA achieved a 51% performance decrease on the denser Cora during an evasion attack with METAATTACK, but only a 19% decrease on the sparser PubMed dataset.

### 4.2 WHICH GRAPH-AWARE PARADIGM IS MORE SUSCEPTIBLE TO GRAPH ADVERSARIAL ATTACKS AND WHY?

Our experiments (Table 1) show that LLAGA is more susceptible to attacks than GRAPH-PROMPTER due to its *sensitivity to structural perturbations*. LLAGA integrates the graph structure in two ways: 1) by generating a fixed-length node-level sequence for each node and 2) by computing a graph Laplacian embedding. Each position in this sequence uniquely corresponds to a relative structural position within the original graph. Since the node embeddings are directly derived from the perturbed graph structure, these attacks significantly influence the information processed by the LLM, leading to performance degradation. In contrast, GRAPHPROMPTER's textual node embeddings remain unchanged during structural perturbations, as only the GNN component is affected. This design allows the unperturbed textual features to augment the perturbed graph structure, enhancing robustness.

### 4.3 NODE SEQUENCE TEMPLATE INJECTION ATTACKS

We demonstrate a new attack surface in LLAGA. Specifically, by exploiting the neighborhood detail template used to construct the node sequences. The attack is illustrated in Figure 2 and the algorithm is in Appendix A.4. To obtain the node sequence for a particular node $u$, the neighborhood detail template in LLAGA first constructs a *fixed-shape* computational tree for each node, where the children of a node are its sampled 1-hop neighbors. This process is repeated to sample neighbors for each subsequent level of the tree. If a node has an insufficient number of neighbors, a placeholder is used to maintain the fixed size of e.g. $k = 10$ children, which is consistent with LLAGA's original design (see "Clean" from Figure 2 and lines 1-14 in the algorithm). This fixed structure provides a new attack surface.

We demonstrate a new attack surface in LLAGA by exploiting its neighborhood detail template algorithm. An attacker can craftily inject malicious nodes as placeholders through graph manipulation

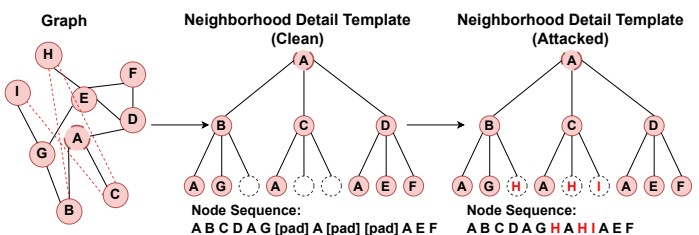

Figure 2: Illustration of the node sequence template injection attacks. The attack shown here is the Non-adjacent Injection attack (NI). The red dashed lines are the edge perturbation by the attacker. For the Supernode Injection (SI) and Multiple Supernode Injection (MSI) attack, the supernode(s) are injected in the place of the placeholders. Here, $k = 3$.

Table 2: Results of the node sequence template injection attack across different datasets and methods. *NI*, *SI*, and *MSI* indicates the non-adjacent, single supernode, and multiple supernodes injection attacks respectively. Clean refers to the performance of the model without any attack.

| Data | Attack | NI | SI | MSI |
|---|---|---|---|---|
| Cora | Clean | | ——— $0.89_{\pm 0.07}$ ——— | |
| | Poisoning | $0.86_{\pm 0.03}$ | $0.84_{\pm 0.06}$ | $0.82_{\pm 0.04}$ |
| | Evasion | $0.42_{\pm 0.08}$ | $0.36_{\pm 0.05}$ | $0.30_{\pm 0.05}$ |
| Citeseer | Clean | | ——— $0.71_{\pm 0.04}$ ——— | |
| | Poisoning | $0.69_{\pm 0.05}$ | $0.66_{\pm 0.03}$ | $0.62_{\pm 0.02}$ |
| | Evasion | $0.36_{\pm 0.06}$ | $0.30_{\pm 0.04}$ | $0.24_{\pm 0.05}$ |
| PubMed | Clean | | ——— $0.90_{\pm 0.04}$ ——— | |
| | Poisoning | $0.88_{\pm 0.04}$ | $0.85_{\pm 0.05}$ | $0.81_{\pm 0.03}$ |
| | Evasion | $0.85_{\pm 0.07}$ | $0.82_{\pm 0.04}$ | $0.76_{\pm 0.03}$ |

to significantly degrade model performance. We propose three strategies for this attack. The first, termed *Non-Adjacent Injection (NI)*, involves injecting nodes that are not within the 2-hop neighborhood of a target node $u$ (see "Attacked" from Figure 2 and lines 21-27). The second approach, referred to as *Supernode Injection (SI)*, involves injecting a high-degree node as a placeholder, ensuring it is outside the target node's 2-hop neighborhood. The third, *Multiple Supernode Injection (MSI)* is a variant of *SI* that injects multiple distinct supernodes as placeholders (lines 35-43). The rationale is that the inclusion of uninformative nodes in the node sequence corrupts the representation of the target node, leading to a poorly trained projection layer. This compromises the integrity of the model's learning and inference process.

**Results.** As shown in Table 2, LLAGA shows high resilience to poisoning attacks on the Cora and Citeseer datasets, but suffers a significant performance drop during evasion, with accuracies decreasing to 0.42, 0.36, and 0.30 for Cora and to 0.36, 0.30, and 0.24 for Citeseer on the *NI*, *SI*, and *MSI* attacks respectively. Conversely, on the larger PubMed dataset, the model demonstrates a stronger resilience to all injection attacks, with only a modest drop in accuracy for evasion (0.85, 0.82, and 0.76). One reason is that in smaller, more tightly-connected graphs like Cora and Citeseer, a localized injection has a more concentrated impact, severely disrupting the model's ability to reason about the local neighborhood. In contrast, on the much larger PubMed graph, the effect of the same attack is diluted, leading to a less significant performance degradation. Compared to traditional graph adversarial methods, node injection attacks are more effective. For instance, the *MSI* method achieved evasion attack accuracies of 0.30, 0.24 and 0.76 on Cora, CiteSeer and PubMed, respectively, while METAATTACK only reached 0.44, 0.50 and 0.73 (Table 1).

## 4.4 POISONING AND EVASION ATTACK VIA FEATURE PERTURBATION

Existing graph adversarial attacks primarily focus on structural perturbations (Li et al., 2023; Wu et al., 2019; Dai et al., 2018; Sun et al., 2022; Zügner & Günnemann, 2019), with the exception of NETTACK (Zügner et al., 2018), which is not transferable to the textual domain. Given that structural attacks are only moderately successful on robust models like GRAPHPROMPTER, and that textual features have proven to be important for model robustness, we explore unnoticeable feature perturbation attacks to degrade model performance. We adopt the imperceptible attack strategies

(a) Performance of feature perturbation attacks, with percentage drop from clean accuracy in parentheses.

| Data | Method | Poisoning | | Evasion | |
|------|--------|-----------|--|---------|--|
| | | LLAGA | GRAPHPROMPTER | LLAGA | GRAPHPROMPTER |
| Cora | Homoglyph | $0.65_{\pm 0.05}$ (27%) | $0.52_{\pm 0.04}$ (13%) | $0.41_{\pm 0.04}$ (54%) | $0.38_{\pm 0.03}$ (37%) |
| | Reordering | $0.58_{\pm 0.06}$ (35%) | $0.45_{\pm 0.05}$ (25%) | $0.24_{\pm 0.05}$ (73%) | $0.20_{\pm 0.05}$ (67%) |
| Citeseer | Homoglyph | $0.54_{\pm 0.04}$ (23%) | $0.65_{\pm 0.04}$ (7%) | $0.32_{\pm 0.03}$ (55%) | $0.52_{\pm 0.05}$ (26%) |
| | Reordering | $0.46_{\pm 0.03}$ (35%) | $0.39_{\pm 0.02}$ (44%) | $0.20_{\pm 0.05}$ (72%) | $0.27_{\pm 0.02}$ (61%) |
| PubMed | Homoglyph | $0.72_{\pm 0.04}$ (20%) | $0.83_{\pm 0.04}$ (8%) | $0.36_{\pm 0.06}$ (60%) | $0.55_{\pm 0.03}$ (39%) |
| | Reordering | $0.67_{\pm 0.05}$ (26%) | $0.80_{\pm 0.03}$ (11%) | $0.28_{\pm 0.04}$ (69%) | $0.42_{\pm 0.04}$ (53%) |

(b) Unified imperceptible attack. Percentage drop from clean accuracy in parentheses.

| Data | Attack | LLAGA | GRAPHPROMPTER |
|------|--------|-------|---------------|
| Cora | Poisoning | $0.35_{\pm 0.05}$ (61%) | $0.28_{\pm 0.08}$ (53%) |
| | Evasion | $0.14_{\pm 0.06}$ (84%) | $0.11_{\pm 0.05}$ (82%) |
| Citeseer | Poisoning | $0.26_{\pm 0.04}$ (63%) | $0.22_{\pm 0.03}$ (69%) |
| | Evasion | $0.10_{\pm 0.09}$ (86%) | $0.13_{\pm 0.07}$ (81%) |
| PubMed | Poisoning | $0.39_{\pm 0.07}$ (57%) | $0.32_{\pm 0.04}$ (64%) |
| | Evasion | $0.12_{\pm 0.05}$ (87%) | $0.19_{\pm 0.04}$ (79%) |

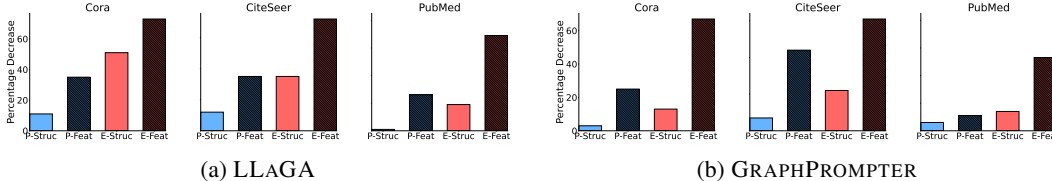

(a) LLAGA      (b) GRAPHPROMPTER

Figure 3: Performance of structure and feature-based adversarial attacks on graph-aware LLMs. The attack success rate here is measured by the percentage decrease (y-axis) to the original accuracy. P-Struc, P-Feat, E-Struc, E-Feat refer to the poisoning and evasion attacks based on structural perturbation and feature perturbation.

proposed by (Boucher et al., 2022) but with a key modification: we use a task-agnostic translation objective (English to French) to generate perturbations. The objective is to modify the input text such that the changes are visually identical but lead to an incorrect translation, thereby corrupting the encoded semantics. The intuition is that if the translation for a given text is wrong, the encoded semantics would also be wrong. We employ two imperceptible attacks: *Homoglyphs*, which replaces characters with visually similar ones and *Reorderings*, which use direction control characters to alter the rendering sequence. Examples of Homoglyphs and Reorderngs are in Appendix A.7.1. Following (Boucher et al., 2022), we use differential evolution (Storn & Price, 1997) to optimize these perturbations, maximizing the Levenshtein distance between the model's output on the perturbed and unperturbed text. The perturbation budget is set to 10% of the average length of each textual feature.

**Results.** As shown in Table 3a, Reordering attacks are generally more potent than Homoglyph attacks across all models and datasets. On Cora and CiteSeer, LLAGA shows vulnerability to both attack types, with poisoning attacks causing accuracy drops of 27% and 23% for Homoglyph and a more significant 35% and 25% for Reordering, respectively. In both cases, GRAPHPROMPTER proves more resilient. The PubMed dataset exhibits similar trends, though the overall impact is less pronounced than on the smaller datasets. A comparison with structural perturbations reveals a fundamental shift in vulnerabilities, as shown in Figure 3. Feature perturbation attacks (specifically Reordering) outperform structural attacks (METAATTACK) by a large margin on both LLAGA and GRAPHPROMPTER. This suggests that graph-aware LLMs place a greater emphasis on input features than on graph structure, making them more vulnerable to feature perturbations. We further explore this by combining both attack types in a unified setting in the next section.

## 4.5 A UNIFIED IMPERCEPTIBLE ATTACK ON GRAPH-AWARE LLMS

We propose a unified attack that combines both feature and structural perturbations to create a more imperceptible and effective adversarial strategy. We hypothesize that combining these attack types will more significantly degrade model accuracy than either method alone, while maintaining imperceptible alterations to the graph structure and features. For the structural perturbation, we use METAATTACK, which modifies graph structure by subtly adjusting edges. For the feature component, we use a combination of Homoglyph and Reordering attacks, as described in previous section. Together, these perturbations create a highly imperceptible attack that simultaneously targets the relational and textual information on which graph-aware LLMs rely.

**Results.** As shown in Table 3b, the unified attack, which jointly applies structural and feature perturbations, is significantly more effective than either perturbation method alone in degrading model performance across various datasets. This analysis offers a key insight into the architecture of graph-

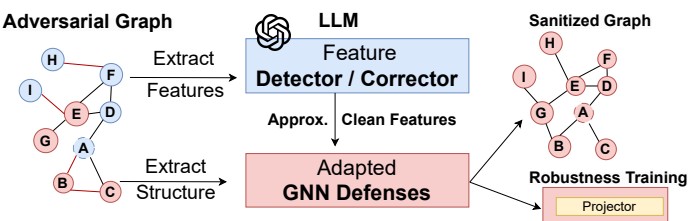

Figure 4: **Overview of GALGUARD defense.** Perturbed node features and edges are colored in blue and red, respectively. The LLM identifies perturbations in node features extracted from an adversarial graph and generates a corrected version. The refined features, together with the perturbed edges, are then passed through existing GNN defenses, which can either be used to obtain a sanitized graph or adapted for projectors' robustness training.

aware LLMs: unlike traditional GNNs, which are primarily impacted by structural perturbations, these models exhibit substantial vulnerability to feature-level attacks. The effectiveness of the unified attack indicates that these models rely on the consistency of both graph structure and node features, making them highly susceptible to such combined, inconspicuous modifications.

### 4.6 ABLATION STUDY

**Varying Feature Perturbation.** To better understand the relationship between perturbation level and attack success, we study the impact of varying the percentage of perturbed features. Our analysis of the Reordering attack, which exhibits similar patterns to the Homoglyph attack, reveals a consistent trend as the perturbation percentage varies from $1\%$ to $10\%$. For all models and datasets (Cora, Citeseer, and PubMed), reducing the percentage of perturbed features leads to a less successful attack, as expected. Notably, as the perturbation level decreases, the success rates of poisoning and evasion attacks tend to converge. This is because with fewer perturbations, the adversarial signal becomes too weak to significantly impact the model, regardless of whether it was introduced during the training (poisoning) or inference (evasion) phase. A detailed plot illustrating these trends is in Figure 6.

**Does the Choice of LLM Affect Attack Performance?** We evaluate whether the foundation LLM influences adversarial robustness in graph-aware LLMs. LLAGA uses Vicuna-7B (Chiang et al., 2023) and GRAPHPROMPTER uses Llama2-7B (Touvron et al., 2023). We replace LLAGA's backbone with Llama2-7B and compare performance under poisoning and evasion attacks on all datasets. Both models achieve similar accuracy in clean settings (e.g. Vicuna: $0.89 \pm 0.07$, Llama2: $0.90 \pm 0.03$ on Cora) and exhibit comparable vulnerability under attacks. The trend holds when swapping LLMs in GRAPHPROMPTER. These results suggest that the choice of foundation LLM has minimal impact on adversarial robustness. The full results are in Table 7 in the Appendix.

## 5 OUR GALGUARD DEFENSE

Existing GNN defenses against adversarial attacks are ineffective for graph-aware LLMs due to their dual vulnerability to structural and feature perturbations. Many defenses rely on node feature similarity to prune or reweight edges Wu et al. (2019); Dai et al. (2018); Zhang & Zitnik (2020); Jin et al. (2020); Zhang & Ma (2024), but they assume access to clean features. When features are perturbed, these methods fail (see Figure 8 in the Appendix). Similarly, defenses that decouple structure and features Shanthamallu et al. (2021); Wu et al. (2021) and even OCR-based sanitization Boucher et al. (2022) break down under coordinated attacks. This dual vulnerability necessitates a new, integrated defense.

To address this, we propose graph-aware LLM defense (GALGUARD), an end-to-end defense strategy combining two complementary approaches: (1) an LLM-based feature corrector and (2) adapted GNN-based structural defenses.

**Feature Robustness Strategy.** As shown in Figure 4, an LLM serves as an adaptive mechanism to analyze textual node features for anomalous patterns. It provides approximate corrections to

Table 4: Performance of METAATTACK and the proposed defense GALGUARD under poisoning and evasion attacks. GALGUARD$_p$ combines feature correction with graph purification.

| Data | Method | LLaGA | | GRAPHPROMPTER | |
|---|---|---|---|---|---|
| | | Poisoning | Evasion | Poisoning | Evasion |
| Cora | METAATTACK | $0.79_{\pm 0.03}$ | $0.44_{\pm 0.06}$ | $0.58_{\pm 0.05}$ | $0.52_{\pm 0.04}$ |
| | GALGUARD$_p$ | $0.82_{\pm 0.05}$ | $0.62_{\pm 0.04}$ | $0.59_{\pm 0.03}$ | $0.56_{\pm 0.04}$ |
| | GALGUARD | $\mathbf{0.85}_{\pm 0.04}$ | $\mathbf{0.83}_{\pm 0.04}$ | $\mathbf{0.60}_{\pm 0.05}$ | $\mathbf{0.59}_{\pm 0.03}$ |
| CiteSeer | MetaAttack | $0.63_{\pm 0.04}$ | $0.50_{\pm 0.03}$ | $0.65_{\pm 0.05}$ | $0.56_{\pm 0.05}$ |
| | GALGUARD$_p$ | $0.64_{\pm 0.03}$ | $0.56_{\pm 0.02}$ | $0.67_{\pm 0.05}$ | $0.60_{\pm 0.06}$ |
| | GALGUARD | $\mathbf{0.67}_{\pm 0.04}$ | $\mathbf{0.62}_{\pm 0.04}$ | $\mathbf{0.68}_{\pm 0.04}$ | $\mathbf{0.64}_{\pm 0.05}$ |
| PubMed | MetaAttack | $0.89_{\pm 0.08}$ | $0.73_{\pm 0.04}$ | $0.85_{\pm 0.02}$ | $0.77_{\pm 0.03}$ |
| | GALGUARD$_p$ | $0.90_{\pm 0.06}$ | $0.78_{\pm 0.03}$ | $0.85_{\pm 0.04}$ | $0.81_{\pm 0.04}$ |
| | GALGUARD | $\mathbf{0.90}_{\pm 0.01}$ | $\mathbf{0.87}_{\pm 0.02}$ | $\mathbf{0.89}_{\pm 0.02}$ | $\mathbf{0.89}_{\pm 0.01}$ |

restore feature integrity, thereby mitigating the effect of feature-level perturbations. This LLM-based correction is a practical solution for feature vulnerabilities. The specific instructions and prompts are in Appendix A.7.2. For our experiments, we employ GPT-4 Turbo as the LLM corrector.

**Graph Structure Robustness Strategy.** To enhance structural robustness, we employ a two-step approach. First, we adopt a *graph purification heuristic* (Wu et al., 2019) as a preprocessing step. For each node $v$, we compute the cosine similarity between its embedding $\phi(x_v)$ and its neighbors' embeddings $\phi(x_u)$ and remove edges with low similarity. Second, we integrate *GNNGuard* Zhang & Zitnik (2020). We directly apply GNNGuard's principles to the message-passing layers of GRAPHPROMPTER. For LLaGA, we propose a novel adaptation: we incorporate the purification step into the construction of the computational tree for its node sequence. Furthermore, we introduce a learnable *global structural context embedding* ($M_{global}$) as a learnable parameter in LLaGA's projector. By concatenating $M_{global}$ with the purified node sequence embedding and jointly training them, the model learns to operate robustly on potentially perturbed graph inputs. The plug-and-play design of GALGUARD's allows other defenses, e.g. Pro-GNN Jin et al. (2020), to be used instead of GNNGuard.

**Results.** Table 4 summarizes the performance, measured in accuracy, of models under METAATTACK, as well as our defense approaches: GALGUARD$_p$, which combines a feature corrector with a graph purification module, and the full GALGUARD, which integrates the feature corrector, the graph purification module, and GNNGuard (or its adaptation). Across all evaluated datasets (Cora, CiteSeer, and PubMed) and for both poisoning and evasion attack types, our defenses consistently mitigate the impact of METAATTACK. Specifically, GALGUARD$_p$ proves to be an effective foundational step, demonstrating that addressing feature perturbations with the LLM corrector and applying initial graph purification significantly improve robustness. Notably, the full GALGUARD consistently yields superior improvements in model robustness compared to GALGUARD$_p$, highlighting the synergistic effect of integrating the GNN-inspired structural defense.

## 6 CONCLUSION

In this paper, we take a substantial first step in uncovering the vulnerabilities of graph-aware LLMs to adversarial attacks, a largely unexplored area. Our work reveals that the methods used for integrating graph data are a critical source of vulnerability. We discovered new attack surfaces, demonstrating that design choices like the node sequence template in LLaGA significantly increase a model's susceptibility. Furthermore, our findings show that feature perturbation attacks are highly effective (unlike traditional GNNs), often outperforming traditional structural attacks, and that a unified attack combining both perturbations can render graph-aware LLMs near-useless.

To address this, we propose an end-to-end defense, GALGUARD, which integrates an LLM-based feature corrector with adapted GNN-based structural defenses to provide robust protection. As these models become more prevalent and has potentials in applications spanning social networks, healthcare, and finance, ensuring their resilience to adversarial exploitation is of the essence. We hope this work serves as a foundation for further research, inspiring deeper exploration into the vulnerabilities and defenses of graph-aware LLMs and fostering both theoretical insights and practical safeguards for their real-world deployment.

**Reproducibility Statement.** To facilitate reproducibility, we provide the full source code for our experiments at `https://anonymous.4open.science/r/ AdvAttackGraphAwareLLMs-C217/` . This includes scripts for data preprocessing, model training, evaluation and defense.

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

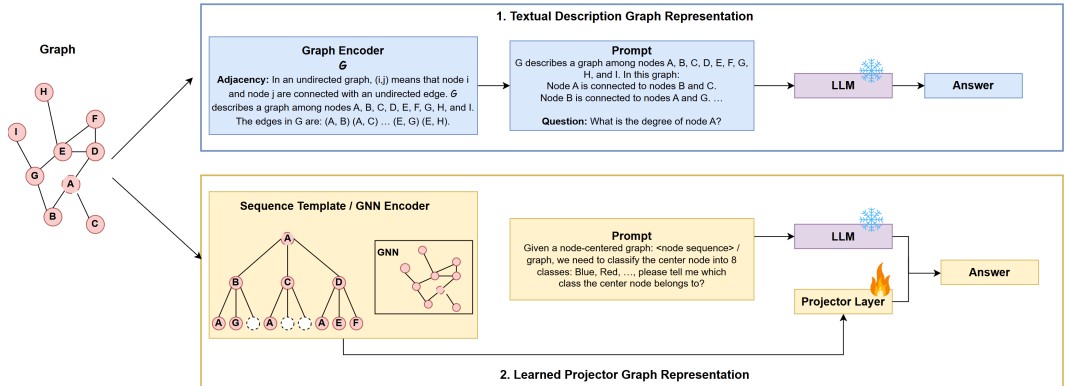

Figure 5: **Graph encoding-based adaptations of LLMs.** In the textual description graph representation, graphs are explained textually (top part). In the learned projector representation, graphs are encoded either by node templates that turns the graph into sequences or by GNNs (bottom part). Note that in both cases, the LLM is frozen. Only the projector that maps the graph into tokens is trained.

# A APPENDIX

**Organization:** The Appendix is organized as follows. Appendix A.1 provides a detailed taxonomy of graph-aware LLMs. We discuss related work on attacks against Graph Neural Networks (GNNs) in Appendix A.2. Appendix A.3 gives an overview of the adversarial attacks, NETTACK and METAATTACK, while Appendix A.4 details the algorithm for the node sequence injection attack on LLAGA. We present the results of our ablation studies in Appendix A.5, including the effect of varying the amount of perturbed features and the choice of base LLM on attack performance. Additional experiment on the ArXiv dataset is in Appendix A.6. Finally, Appendix A.7 shows examples of feature attacks and the template for the LLM corrector, and Appendix A.8 discusses the implications of our research.

Table 5: Dataset statistics.

| Data | #Node | #Edge | Sparsity (‰) |
|---|---|---|---|
| Cora | 2708 | 5429 | 14.81 |
| CiteSeer | 3327 | 4614 | 8.34 |
| PubMed | 19717 | 44338 | 2.28 |

## A.1 GRAPH ENCODING-BASED TAXONOMY OF LLM ADAPTATIONS

The integration of large language models (LLMs) with graph data has been an emerging area of research, with various approaches exploring how LLMs can be adapted or enhanced for graph-related tasks such as reasoning, representation, and inference. These approaches can be largely divided into two categories based on their methods for encoding graphs for LLMs: ***textual descriptions of graphs***, where graph information is encoded as natural language, and ***learned graph projector***, where a projector is utilized to encode graph data into embeddings or structured forms that LLMs can directly process. In this section, we describe the methods used in the different encoding approaches for integrating graph data into LLMs. Figure 5 provides the illustration of both approaches and Table 6 provides a summary of the differences between both approaches.

### A.1.1 GRAPH REPRESENTATION AS A TEXTUAL DESCRIPTION

In this first category, graph data is represented as natural language, allowing LLMs to process graph structures as textual descriptions. This approach has been explored extensively with works such as (Guo et al., 2023; Liu & Wu, 2023; Ye et al., 2023; Wenkel et al., 2023; Wei et al., 2024b; Fatemi et al., 2023; Tan et al., 2023; Zhao et al., 2023; Wei et al., 2024a) converting graph data

into textual formats, often paired with specific instructions to query LLMs. For example, Tan et al. (2023) introduced a unified framework for attributed graph embedding by generating random walk-based textual sequences from graphs, which are then used to fine-tune LLMs. Similarly, Zhao et al. (2023) focused on graph reasoning within text spaces, translating graph structures into natural language to facilitate LLM interaction. Extending this idea, Wei et al. (2024a) augmented textual representations of graphs with visual representations of the graph, enhancing the reasoning capabilities of LLMs on graph data. In multimodal applications, Liu et al. (2023) and Su et al. (2022) linked molecule graphs with textual descriptions, enabling advances in drug discovery and molecular retrieval. Furthermore, Huang et al. (2023a) investigated how incorporating graph structure into LLM prompts can enhance predictive performance. Their findings revealed that LLMs tend to process graph-related prompts as contextual paragraphs, rather than explicitly recognizing graph structures. This further underscores the importance of prompt design, as seen in studies like (Brannon et al., 2023; Zhang et al., 2023; Sun et al., 2023; Bi et al., 2024; Pan et al., 2024), which extended LLMs to heterogeneous and large-scale graph tasks. Complementing these efforts, (He et al., 2024) explored using LLM-generated explanations as features to enhance both the performance and interpretability of LLMs on graphs.

While still representing graphs as textual description, another line of work integrates pretraining strategies with GNNs to enhance the LLMs' graph reasoning capabilities. For instance, Chen et al. (2023a) proposed two LLM-GNN integration approaches: 1) using LLMs to enhance node features for node classification, and 2) employing LLMs as direct predictors for graph tasks. In line with this, Chandra et al. (2020) utilized LLMs as text encoders and GNNs as graph structure encoders for tasks such as fake news detection in online communities. Similarly, Chen et al. (2023b) explored the possibility of performing node classification without explicit labels by using LLMs to annotate nodes, which are then used in training a GNN model. Likewise, Mavromatis et al. (2023) utilized LLM as a feature extractor for GNNs.

For inference, Zhu et al. (2024) proposed a fine-tuning method for textual graph that combined LLMs and GNNs through a tunable side structure alongside each layer of the LLM, enhancing both training efficiency and inference speed. Similarly, Duan et al. (2023) introduced SimTeG, a simple yet effective approach that enhances textual graph learning by leveraging LLMs. In their method, the LLM is first fine-tuned on a downstream task, such as node classification. Afterward, node embeddings are generated by extracting the last hidden states from the fine-tuned LLM. Additionally, Zhang (2023) proposed Graph-toolformer, which leverages prompt engineering, powered by ChatGPT, to augment LLMs' graph reasoning capabilities.

While these approaches make LLMs more accessible for graph-related tasks, they often struggle to fully capture the structural intricacies inherent in graph data. Moreover, the performance of these methods remains heavily dependent on effective prompt engineering, which can be a challenging process, requiring careful design and experimentation to obtain optimal results.

### A.1.2 GRAPH REPRESENTATION AS A LEARNED PROJECTOR

To address the challenges posed by textual representation approaches, learned projector methods offer an alternative by transforming graph data into embeddings or graph vectors that LLMs can process more directly. For instance, Chen et al. (2024) introduced LLAGA, a large language and graph assistant that retains the general-purpose nature of LLMs while transforming graph structures into graph tokens, a format suitable for LLM processing. Similarly, (Liu et al., 2024; Qin et al., 2023) incorporated graph structure information into a pre-trained LLM through a tailored disentangled graph neural network layers. Tian et al. (2024) proposed using graph neural prompts to encode knowledge graph, transforming them into graph embeddings that LLMs can leverage during inference. Furthermore, Huang et al. (2023b) introduced a prompt-based node feature extractor (G-Prompt) for few-shot learning on text-attributed graphs. Their method combines a graph adapter (a learnable GNN layer) with task-specific prompts to capture neighborhood information and extract informative node features.

Another promising line of work condenses graph information into fixed-length prefixes using graph transformers, which are attached to each layer of the LLM to enhance its reasoning capabilities (Chai et al., 2023). Similarly, Yang et al. (2021) developed GraphFormers which seamlessly integrates text encoding and graph aggregation in an iterative workflow by nesting GNNs alongside the transformer blocks of a language model. Additionally, Peng et al. (2024) developed ChatGraph, an interactive

Table 6: Comparison of graph encoding-based adaptations of LLMs.

| Aspect | Textual Description Approach | Learned Projector Approach |
|---|---|---|
| **Graph Encoding** | Converts graph data (nodes, edges) into natural language descriptions. | Uses a learned projector to encode graph structures into embeddings or graph tokens. |
| **LLM Modifications** | No modification to the LLM. | No modification to the LLM. |
| **Graph Structure Representation** | Describes graph topology and relationships using textual narratives. | Encodes relational and structural information directly into vector representations. |
| **Flexibility** | Can only be used for simple graphs. | More flexible in representing complex graph structures. |
| **Computational Cost** | No additional training is required. Lower cost. | Lightweight projector is trained, typically a linear layer. Minimal cost. |
| **Suitability for Graph Tasks** | May lose some relational nuance due to limitations in textual descriptions. | Captures detailed structural information, improving performance on graph-specific tasks. |

interface that allows users to interact with their graphs through natural language. Here, graphs are represented as sequential paths and the LLM is fine-tuned on the graph data to support graph analysis.

In this paper, we do not consider projector adaptations that require retraining the LLM, as they significantly increase computational costs without yielding substantial performance gains. Instead, we focus on adaptation methods that utilize **frozen pre-trained LLMs** (Chen et al., 2024; Liu et al., 2024; Qin et al., 2023; Tian et al., 2024; Huang et al., 2023b). This approach not only reduces computational costs but also offers greater flexibility in integrating with various LLM models. Additionally, we can take advantage of LLMs' extensive knowledge and capabilities without the overhead of retraining while still adapting them for specific graph tasks. Throughout this work, we employ two representative projector methods: **LLAGA** (Chen et al., 2024), which utilizes node-level templates and a linear projector, and **GRAPHPROMPTER** (Liu et al., 2024), which employs a GNN-based projector to encode the graph.

**This work.** In this work, we do not consider textual description methods, as they often struggle to fully capture the structural intricacies inherent in graph data. These methods are generally limited to simpler tasks, such as describing basic graph properties like node degree, and become infeasible for larger graphs due to scalability issues. In particular, the token limit in prompts restricts the detailed descriptions required for larger graph structures, quickly exceeding allowable token counts. Additionally, from a vulnerability perspective, textual description methods do not introduce any unique attack surface; since the graph is represented as simple text descriptions (e.g., "Node A is connected to Node B"), they are susceptible to traditional text-based adversarial attacks rather than the structural or feature perturbations specific to graph-based models. Therefore, they lack the complexity that would expose them to novel, graph-specific vulnerabilities, making them less relevant for our investigation into adversarial robustness of graph-aware LLMs.

## A.2 ATTACKS ON GNN

Several types of attacks exist on GNNs, including membership inference attacks (Olatunji et al., 2021; He et al., 2021; Conti et al., 2022), which aim to determine whether a specific node was part of the model's training set; attribute inference attacks (Gong & Liu, 2018; Olatunji et al., 2023a; Duddu et al., 2020), which exploit the GNN's output and learned embeddings to infer sensitive or missing node attributes; property inference attacks (Zhang et al., 2022; Wang & Wang, 2022), which seek to extract global properties of the entire graph; adversarial attacks (Zügner et al., 2018; Zügner & Günnemann, 2019; Li et al., 2023; Dai et al., 2018; Sun et al., 2022), which attempt to degrade the model's performance by introducing subtle perturbations to the graph or node features;

and graph reconstruction attacks (Zhang et al., 2021; Olatunji et al., 2023b), which aim to reconstruct the underlying graph structure using the learned embeddings or model outputs.

Among these attack types, adversarial attacks are of particular concern due to their direct impact on the model's performance. In adversarial attacks, small, carefully crafted *perturbations* are introduced into the graph, such as modifying node features or altering edges, with the goal of significantly degrading the GNN's performance. These attacks can be categorized into two types: **poisoning attacks**, which occur during the *training phase* by manipulating the training data, and **evasion attacks**, which occur at *test time*, where the adversary introduces perturbations to the input graph to mislead the model's predictions. For instance, Zügner et al. (2018) demonstrated that adversarial perturbations on graph structures or node features can significantly degrade the performance of GNNs. Their method, NETTACK, employed a greedy optimization technique to compute the minimal perturbations that would maximize the model's loss function during training. This approach effectively identifies the smallest changes in the features or graph structure needed to disrupt the model's predictions. Li et al. (2023) revisited the problem of adversarial attacks on graph data from a data distribution perspective, highlighting how adversarial examples can significantly alter the underlying data distribution and proposed a novel defense mechanism to mitigate such effects. Similarly, Wu et al. (2019) provided a comprehensive analysis of adversarial examples on graph data, offering deep insights into both attacks and defenses. They focused on the vulnerabilities of GNNs to small perturbations and proposed defense techniques designed to detect and counteract adversarial modifications, thus enhancing model robustness. METAATTACK (Zügner & Günnemann, 2019) performed both poison and evasion attacks on GNNs by treating the graph structure as a hyperparameter that can be optimized. The attack directly modifies the graph structure using gradient descent, formulating the problem as a bi-level optimization task. Their method utilized meta-gradients, which capture how small perturbations in the graph structure affect the attacker's loss after training, enabling precise updates to the graph that degrade the model's performance.

In this paper, we leverage **NETTACK** (Zügner et al., 2018) and **METAATTACK** (Zügner & Günnemann, 2019) as representative adversarial attack methods, encompassing both poisoning and evasion strategies, to conduct our investigation. Specifically, poisoning attacks target model integrity during training, while evasion attacks compromise performance at inference or test time.

## A.3 HOW NETTACK AND METAATTACK WORK

**NETTACK** performs graph structure perturbation by adding and removing edges between nodes, while ensuring that the degree of the perturbed graph is preserved. It uses a greedy optimization technique to compute the minimal perturbations that will maximize the model's loss function during training, with respect to a surrogate model. This surrogate model is a linearized 2-layer graph convolutional network (GCN), which approximates how changes in the graph structure affect the classification result. The primary objective of NETTACK is to increase the model's confidence in incorrect classifications by maximizing the distance between the logits of the perturbed graph and those of the clean graph. To achieve this, NETTACK identifies candidate edges for perturbation around a target node and evaluates a score function, for each connected edge. This score indicates how significantly altering a particular edge will impact the classification result of the target node. The edge with the highest score is then selected, and the graph is updated by either adding or removing that edge. Since NETTACK is a targeted attack (only cause misclassification of a specific single node), we adapt the same method in (Zügner & Günnemann, 2019) to convert it into untargeted attack (compromise node classification performance of the model) which randomly select one test node and attack it using NETTACK while considering all nodes in the network. We then aggregate the result over the test nodes.

**METAATTACK** optimizes the graph structure directly via gradient descent by treating the adjacency matrix (graph structure) as a hyperparameter. It frames the attack as a bi-level optimization problem: the inner level minimizes the training loss on the original graph, while the outer level minimizes the loss on the poisoned graph. To achieve this, METAATTACK utilizes a surrogate model, specifically a linearized two-layer GCN, to approximate how changes in the graph structure affect model performance. A score function evaluates the impact of each perturbation, and the attack flips the sign of meta-gradients (gradient wrt hyperparameters) for connected nodes, indicating which edges to remove. The algorithm then greedily selects perturbations—either adding or removing edge based on those with the highest scores, to maximize the attack's impact.

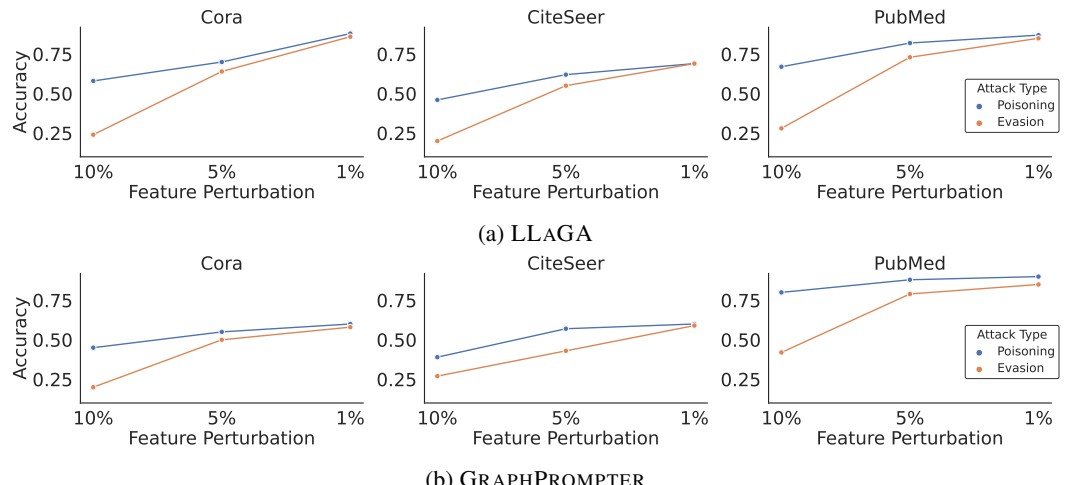

(a) LLAGA

(b) GRAPHPROMPTER

Figure 6: Performance of the Graph-aware LLM when the percentage of the perturbed feature is varied. The lower the accuracy, the better.

For both methods, we use the same unoticeability constraint introduced by (Zügner et al., 2018), which ensures that the graph's degree distribution only changes slightly. Therefore, we only perturb 10% of the edges.

In our poisoning attack experiments with LLAGA, we employ the perturbed graph structure obtained from NETTACK or METAATTACK to generate the node-level sequences and retrain the projector that converts the node-level sequences into a sequence of token embeddings. Similarly, for GRAPH-PROMPTER, we utilize the perturbed graph structure, re-encode the graph using a graph attention network (GAT), and retrain the projector with the perturbed graph. For evasion attacks, the graph-aware LLM is trained on unperturbed data, with perturbations introduced only during the inference stage.

### A.4 ALGORITHM FOR NODE SEQUENCE TEMPLATE INJECTION ATTACKS

We present the algorithm for the node sequence template injection attack in LLAGA in Algorithm 1. The model constructs a *fixed-shape* computational tree for each node $u$, sampling $k$ neighbors per node up to depth $d$ (lines 1–14). When a node has fewer than $k$ neighbors, placeholders are inserted, creating an attack surface. We exploit this by injecting malicious nodes into placeholder positions through graph manipulation. We propose three strategies: *Non-Adjacent Injection (NI)* inserts nodes outside the 2-hop neighborhood of u (lines 21–27); *Supernode Injection (SI)* injects a high-degree node into one placeholder (lines 29–33); and *Multiple Supernode Injection (MSI)* distributes distinct high-degree nodes across all placeholders (lines 35–43). These uninformative nodes corrupt the node sequence, degrading the projection layer's performance. The inclusion of uninformative nodes in the sequence corrupts the target's representation, impairing training of the projection layer.

### A.5 ABLATION STUDIES

Here, we present the results for the ablation studies presented in the main paper.

### A.5.1 ABLATION RESULTS: VARYING FEATURE PERTURBATION

In Figure 6, we present the performance of two graph-aware LLMs, LLAGA and GRAPHPROMPTER, as the percentage of perturbed features is varied. We observe that increasing the proportion of perturbed features consistently degrades model accuracy for both models, indicating their sensitivity to feature-level attacks.

---

**Algorithm 1** Injection Attacks on Neighborhood Detail Template for Obtaining Node Sequence.

---

**Input:** Graph $G = (V, E)$, target node $u$, tree depth $d$, children size $k$
**Output:** Modified graph $G'$ with injected nodes
1: **procedure** NEIGHBORHOODDETAILTEMPLATE($u, d, k$)
2:      $T \leftarrow \emptyset$                                        ▷ Initialize empty tree
3:      $T.\text{root} \leftarrow u$
4:      **for** level $\ell = 1$ to $d$ **do**
5:          **for** each node $v$ at level $\ell - 1$ **do**
6:              $N_v \leftarrow \text{Sample}(\mathcal{N}(v), k)$                  ▷ Sample $k$ neighbors
7:              **if** $|N_v| < k$ **then**
8:                  Fill remaining slots with placeholders
9:              **end if**
10:             Add $N_v$ as children of $v$ in $T$
11:          **end for**
12:      **end for**
13:      **return** $T$
14: **end procedure**

15: **procedure** ATTACK($G, u, d, k$, AttackStrategy)
16:      $G' \leftarrow G$                                    ▷ Create copy of original graph
17:      $T \leftarrow$ NEIGHBORHOODDETAILTEMPLATE($u, d, k$)
18:      $P \leftarrow \text{GetPlaceholders}(T)$
19:      **if** $|P| > 0$ **then**
20:          $\mathcal{N}_2 \leftarrow$ 2-hop neighborhood of $u$
21:          **if** AttackStrategy = NI **then**
22:              $V_{\text{non-adjacent}} \leftarrow V \setminus \mathcal{N}_2$
23:              $V_{\text{inject}} \leftarrow \text{Sample}(V_{\text{non-adjacent}}, |P|)$
24:              **for** each placeholder position $p \in P$ **do**
25:                  $v \leftarrow$ Next node from $V_{\text{inject}}$
26:                  Add edge $(parent(p), v)$ to $G'$
27:              **end for**
28:          **else if** AttackStrategy = SI **then**
29:              $V_{\text{non-adjacent}} \leftarrow V \setminus \mathcal{N}_2$
30:              $s \leftarrow \text{GetHighestDegreeNode}(V_{\text{non-adjacent}})$
31:              **for** first placeholder position $p \in P$ **do**
32:                  Add edge $(parent(p), s)$ to $G'$
33:                  **break**                  ▷ Only use first placeholder
34:              **end for**
35:          **else if** AttackStrategy = MSI **then**
36:              $V_{\text{non-adjacent}} \leftarrow V \setminus \mathcal{N}_2$
37:              $S \leftarrow \text{GetNodesOrderedByDegree}(V_{\text{non-adjacent}})$
38:              **for** $i = 1$ to $|P|$ **do**
39:                  $s \leftarrow$ Get $i$-th node from $S$
40:                  $p \leftarrow$ Get $i$-th placeholder from $P$
41:                  Add edge $(parent(p), s)$ to $G'$
42:              **end for**
43:          **end if**
44:      **end if**
45:      **return** $G'$
46: **end procedure**

---

### A.5.2   ABLATION RESULTS: DOES THE CHOICE OF LLM AFFECT ATTACK PERFORMANCE?

The results in Table 7 show that the choice of base LLM has minimal impact on both clean performance and adversarial vulnerability in the LLAGA framework. Across Cora, Citeseer, and PubMed, models built on Vicuna-7B and Llama2-7B achieve nearly identical accuracy under clean conditions and exhibit similar degradation under both poisoning and evasion attacks. For instance, under METAATTACK evasion, Cora performance drops to $0.44 \pm 0.06$ for Vicuna-7B and $0.43 \pm 0.08$ for

Table 7: Impact of base model on model's performance and under different attacks. We show the result for LLAGA. The results demonstrate that the choice of base model does not significantly affect susceptibility to attacks.

| Data | Base Model | Clean | Poisoning | | | Evasion | | |
|------|-----------|-------|-----------|---|---|---------|---|---|
| | | | NETTACK | METAATTACK | MSI | NETTACK | METAATTACK | MSI |
| Cora | Vicuna-7B | $0.89 \pm 0.07$ | $0.87 \pm 0.04$ | $0.79 \pm 0.03$ | $0.82 \pm 0.04$ | $0.55 \pm 0.09$ | $0.44 \pm 0.06$ | $0.30 \pm 0.05$ |
| | Llama2-7B | $0.90 \pm 0.03$ | $0.86 \pm 0.02$ | $0.77 \pm 0.05$ | $0.82 \pm 0.02$ | $0.56 \pm 0.07$ | $0.43 \pm 0.08$ | $0.31 \pm 0.01$ |
| Citeseer | Vicuna-7B | $0.71 \pm 0.04$ | $0.64 \pm 0.05$ | $0.63 \pm 0.04$ | $0.62 \pm 0.02$ | $0.59 \pm 0.04$ | $0.50 \pm 0.03$ | $0.24 \pm 0.05$ |
| | Llama2-7B | $0.71 \pm 0.05$ | $0.65 \pm 0.03$ | $0.60 \pm 0.03$ | $0.61 \pm 0.04$ | $0.58 \pm 0.03$ | $0.48 \pm 0.05$ | $0.26 \pm 0.03$ |
| PubMed | Vicuna-7B | $0.90 \pm 0.04$ | $0.89 \pm 0.03$ | $0.89 \pm 0.08$ | $0.81 \pm 0.03$ | $0.84 \pm 0.05$ | $0.73 \pm 0.04$ | $0.76 \pm 0.03$ |
| | Llama2-7B | $0.91 \pm 0.06$ | $0.88 \pm 0.04$ | $0.90 \pm 0.02$ | $0.80 \pm 0.04$ | $0.85 \pm 0.03$ | $0.75 \pm 0.02$ | $0.77 \pm 0.02$ |

Table 8: Results of the node sequence template injection attack across different datasets and methods. *NI*, *SI*, and *MSI* indicates the non-adjacent, single supernode, and multiple supernodes injection attacks respectively. Clean refers to the performance of the model without any attack.

| Data | Attack | NI | SI | MSI |
|------|--------|-----|-----|-----|
| Cora | Clean | | —— $0.89_{\pm 0.07}$ —— | |
| | Poisoning | $0.86_{\pm 0.03}$ | $0.84_{\pm 0.06}$ | $0.82_{\pm 0.04}$ |
| | Evasion | $0.42_{\pm 0.08}$ | $0.36_{\pm 0.05}$ | $0.30_{\pm 0.05}$ |
| Citeseer | Clean | | —— $0.71_{\pm 0.04}$ —— | |
| | Poisoning | $0.69_{\pm 0.05}$ | $0.66_{\pm 0.03}$ | $0.62_{\pm 0.02}$ |
| | Evasion | $0.36_{\pm 0.06}$ | $0.30_{\pm 0.04}$ | $0.24_{\pm 0.05}$ |
| PubMed | Clean | | —— $0.90_{\pm 0.04}$ —— | |
| | Poisoning | $0.88_{\pm 0.04}$ | $0.85_{\pm 0.05}$ | $0.81_{\pm 0.03}$ |
| | Evasion | $0.85_{\pm 0.07}$ | $0.82_{\pm 0.04}$ | $0.76_{\pm 0.03}$ |
| ArXiv | Clean | | —— $0.73_{\pm 0.02}$ —— | |
| | Poisoning | $0.50_{\pm 0.03}$ | $0.43_{\pm 0.04}$ | $0.42_{\pm 0.03}$ |
| | Evasion | $0.71_{\pm 0.06}$ | $0.69_{\pm 0.02}$ | $0.60_{\pm 0.03}$ |

Llama2-7B. These differences are well within standard deviation. Similar patterns hold across all datasets and attack types. This indicates that attack effectiveness is largely independent of the base LLM.

## A.6 RESULTS ON THE ARXIV DATASET

We now evaluate our attack on the large-scale ArXiv dataset, which consists of 169,343 nodes, 1,166,243 edges, and has a sparsity of 0.81. This experiment aims to investigate the effectiveness of node sequence template attacks in large, real-world graphs. As shown in Table 8, we observe a significant gap between the success rates of poisoning and evasion attacks, with poisoning attacks consistently outperforming evasion across all three injection strategies, which is not the case for the other datasets, Cora, Citeseer and PubMed. This suggests that ArXiv may have inherent structural differences that make it more vulnerable to poisoning yet more robust to evasion. While the sparsity of the ArXiv dataset is very low, inserting a single supernode could significantly affect the model performance.

## A.7 FEATURE ATTACK EXAMPLES AND DEFENSE TEMPLATE

### A.7.1 FEATURE ATTACK EXAMPLES

The examples in the prompt, as shown in Appendix A.7.2, illustrate our adversarial attacks on real features from the Cora dataset. The "input" is the text received by the LLM where the attack is being applied. It is important to note that while a human reader perceives the original text and the perturbed text as visually identical or very similar, the LLM processes a sequence of different Unicode characters or a different character order, which can alter its behavior. We show this in Table 9 for homoglyph and Table 10 for reordering attack.

Table 9: An illustration of the imperceptible homoglyph attack. While the perturbed text is processed as distinct characters by the LLM, the visual output is nearly identical for a human, as shown in the image.

| Original Feature | What LLM Sees | What a Human Sees |
|---|---|---|
| Title: The megaprior heuristic for discovering protein sequence patterns
Abstract: Several computer algorithms for discovering patterns in groups of protein sequences are in use that are based on fitting the parameters of a statistical model to a group of related sequences. | `Title: The mgprir huristic fr discvring prtin squnc pttrns Abstract: Svrl cmputr lgrithms fr discvring pttrns in grups f prtin squncs r in us tht r bsd on fitting th prmtrs f sttisticl mdl to grup of rltd squncs.` | `Title: The megaprior heuristic for discovering protein sequence patterns. Abstract: Several computer algorithms for discovering patterns in groups of protein sequences are in use that are based on fitting the parameters of a statistical model to a group of related sequences.` |

Table 10: An illustration of the imperceptible reordering attack. While the perturbed text is processed as distinct characters by the LLM, the visual output is nearly identical for a human, as shown in the image.

| Original Feature | What LLM Sees | What a Human Sees |
|---|---|---|
| Title: Applications of machine learning: a medical follow up study.
Abstract: This paper describes preliminary work that aims to apply some learning strategies to a medical follow-up study. An investigation of the application of three machine learning algorithms-1R, FOIL and InductH to identify risk factors that govern the colposuspension cure rate has been made. | `Title: Appliatcions of machine lreanign: a medical foall-up study Abstract: This ppaer dsceribes pmerlinary work that aims to aplpy smoe learning sttraegies to a meidcal flolow-up study. An inestivagiotn of the apclpiation of three machine larening algorithms-1R, FOIL and InductH to ientify risk frcatos that govern the clopsuspension cure rtae has been mdae.` | `Title: Applications of machine learning: a medical follow up study Abstract: This paper describes preliminary work that aims to apply some learning strategies to a medical follow-up study. An investigation of the application of three machine learning algorithms-1R, FOIL and InductH to identify risk factors that govern the colposuspension cure rate has been made.` |

### A.7.2 FEATURE CORRECTOR DEFENSE TEMPLATE

We designed a feature corrector module to mitigate the effects of imperceptible attacks, specifically homoglyph substitutions and character reorderings. This module is implemented as a prompt-based defense using LLMs. The prompt is carefully crafted to instruct the LLM to act as a linguistic validator, focusing on detecting and correcting subtle, human-imperceptible perturbations.

The template for our prompt is shown in Figure 7. It explicitly states the task and provides a clear output format, which significantly improves the LLM's performance.

### A.8 DETAILED DISCUSSION

The findings of our work raise several critical insights into the vulnerabilities and design considerations of graph-aware LLMs. As the integration of graph-structured data into LLMs continues to gain traction for graph tasks, it is important to explore not only the performance benefits but also the vulnerabilities of these novel architectures. Our systematic investigation of adversarial attacks on two representative graph-aware LLMs, LLaGA and GRAPHPROMPTER —reveals significant weaknesses, particularly when exposed to poisoning and evasion attacks, which are traditionally used to undermine GNNs. Our key observations are discussed below.

#### A.8.1 IMPACT OF GRAPH ENCODING ON ADVERSARIAL VULNERABILITY

The key observation from our results is that the integration method of graph structure into LLMs can inadvertently increase their susceptibility to adversarial attacks. Specifically, the node sequence template employed in LLAGA appears to create a vulnerability that can be exploited by adversaries, leading to substantial degradation in model performance. This attack surface, unique to LLAGA, highlights a critical tradeoff in graph-aware LLM design: while the use of node-level templates provides an effective means of incorporating graph structure and achieves better performance on graph task, it also exposes the model to new attack vectors not present in traditional GNNs. On the other hand, GRAPHPROMPTER, which uses a GNN-based projector for graph encoding, demonstrates comparatively greater resilience to these attacks, underscoring the potential advantages of hybrid architectures that leverage the strengths of both LLMs and GNNs.

#### A.8.2 SENSITIVITY OF GRAPH-AWARE LLMS TO FEATURE PERTURBATIONS

In addition to the vulnerabilities in the graph encoding techniques, our results also highlight the effectiveness of feature perturbation attacks across both graph-aware LLMs. Despite differences in how graph structure is encoded, both LLAGA and GRAPHPROMPTER exhibit significant sensitivity to imperceptible perturbations in the node features. This highlights a broader trend observed in the LLM paradigm: while these models excel in handling textual data, their robustness to adversarial perturbations, especially those targeting node features, remains a critical challenge. The ability of adversaries to degrade model performance by subtly altering node attributes without being detected calls into question the integrity of graph-aware LLMs, especially in security-sensitive applications. While text sanitization techniques, such as those discussed by Boucher et al. (2022), offer potential defenses, addressing these imperceptible attacks such as homoglyph-based manipulations remains a complex challenge. As proposed, using LLMs as a corrector is one way to mitigate such vulnerabilities, by prompting the model to detect and repair suspicious or malformed inputs before downstream processing.

#### A.8.3 DATA-DEPENDENT VARIABILITY OF POISONING AND EVASION ATTACKS ON GRAPH-AWARE LLMS

Our findings reveal that the efficacy of poisoning and evasion attacks in the context of graph-aware LLMs is not uniform across datasets. While attacks are particularly effective on the Cora, Citeseer and PubMed datasets, we observe a more nuanced interaction between the attack strategies and the graph structure on the ArXiv dataset. This variation emphasizes the importance of dataset characteristics in evaluating the vulnerabilities of graph-aware LLMs. As these models are increasingly deployed in diverse real-world settings, their vulnerabilities must be assessed not only in terms of algorithmic design but also in relation to the specific domains and data they are applied to.

---

**Prompt for Feature Corrector**

You are a linguistic validator. Your task is to detect and correct any subtle, imperceptible text perturbations such as homoglyph substitutions (e.g., replacing "o" with "") and character reorderings. These changes are often used to attack machine learning models.

Given a text string, carefully analyze it. If you identify any perturbations, provide the corrected, original-looking version of the text. If no perturbations are found, return the original text unchanged.

**Example 1 (Homoglyph Substitution):**
**Input:** `Title: The mgprir huristic fr discvring prtin squnc pttrns Abstract: Svrl cmputr lgrithms fr discvring pttrns in grups f prtin squncs r in us tht r bsd on fitting th prmtrs f sttisticl mdl to grup of rltd squncs.`
**Output:** `Title: The megaprior heuristic for discovering protein sequence patterns Abstract: Several computer algorithms for discovering patterns in groups of protein sequences are in use that are based on fitting the parameters of a statistical model to a group of related sequences.`

**Example 2 (Character Reordering):**
**Input:** `Title: Appliatcions of machine lreanign: a medical foall-up study Abstract: This ppaer dsceribes pmerlinary work that aims to aplpy smoe learning sttraegies to a meidcal fllow-up study. An inestivagiotn of the apclpiation of three machine larening algorithms-1R, FOIL and InductH to ientify risk frcatos that govern the clopsuspension cure rtae has been mdae.`
**Output:** `Title: Applications of machine learning: a medical follow up study Abstract: This paper describes preliminary work that aims to apply some learning strategies to a medical follow-up study. An investigation of the application of three machine learning algorithms-1R, FOIL and InductH to identify risk factors that govern the colposuspension cure rate has been made.`

**Example 3 (Mixed Homoglyph & Reordering):**
**Input:** `Title: Submitted to NIPS96, Sctin: Applcations. Prfrnc: Oral prsentatin Reinforcement Lrning for Dynmic Channl Alloction in Abstract: In cllulr tlphne systms, n imrtnt prblm is to dynmiclly alloct th cmmunictin rsurc (chnnls) so as to mximize srvic in stchstic cllr nvirnmnt. This prblem is naturally frmultd s dynmic prgrmming prblm and we us a reinforcement lerning (RL) mthd to find dynmic chnnl llction plicies that ar bttr than prvius heuristic slutins.`
`Output: Title: Submitted to NIPS96, Section: Applications. Preference: Oral presentation Reinforcement Learning for Dynamic Channel Allocation in Abstract: In cellular telephone systems, an important problem is to dynamically allocate the communication resource (channels) so as to maximize service in a stochastic caller environment. This problem is naturally formulated as a dynamic programming problem and we use a reinforcement learning (RL) method to find dynamic channel allocation policies that are better than previous heuristic solutions.`

**Your Task:** Text to correct: {perturbed_text_here}

Corrected Text:

---

Figure 7: The prompt template used for the LLM-based feature corrector module. It explicitly defines the role, task, and expected output format for the LLM.

### A.8.4 IMPLICATIONS FOR GRAPH-AWARE LLM DEVELOPMENT

The vulnerabilities exposed in this study have broader implications for the future development and deployment of graph-aware LLMs. Given their increasing adoption for tasks that were traditionally the domain of GNNs, it is crucial to consider the tradeoffs between performance and vulnerabilities in

the design of these paradigms. As graph-aware LLMs continue to evolve, there is a pressing need for the development of more robust encoding techniques and defenses against adversarial manipulation, particularly in the context of feature perturbations. Furthermore, understanding how adversarial attacks transfer from GNNs to graph-aware LLMs opens new avenues for research in protecting these models against sophisticated adversaries.

### A.8.5 DIFFICULTY IN APPLYING EXISTING GNN DEFENSES AGAINST ADVERSARIAL ATTACKS ON GRAPH-AWARE LLMS

**Defenses on adversarial attacks based on feature perturbation.** Adversarial attacks on GNNs have predominantly focused on structural perturbations, as feature-based perturbations typically fail to influence the predicted class of the target node Dai et al. (2018); Wu et al. (2019); Zügner & Günnemann (2019), making them ineffective attack strategies. Consequently, there are ***no known defenses against feature-based attacks on GNNs***, given their limited effectiveness. However, as demonstrated in the main paper, attacks based on feature perturbations proves effective in the graph-aware LLM paradigm. This introduces a novel challenge within the graph-aware LLM framework, where feature-based attacks, previously ineffective against GNNs, can now successfully manipulate model behavior. This shift highlights the need for developing new defenses that address both structural and feature-based adversarial perturbations. While text sanitization techniques, such as the use of optical character recognition (OCR) as a pre-processsing step for encoding text for the LLM, offer potential defenses, addressing these imperceptible attacks such as homoglyph-based manipulations remains a complex challenge as OCR is imperfect and tends to misinterpret homoglyph at a higher rate than unperturbed text Boucher et al. (2022). This led to our design of self-correcting mechanism for feature perturbation using LLMs.

**Defenses on adversarial attacks based on structural perturbation.** Due to the transferability of structure-based adversarial attacks from GNNs to graph-aware LLMs, particularly in evasion attacks, it is natural to expect that defenses developed for GNNs would also be effective in this paradigm. However, this assumption does not hold. Unlike traditional GNN settings, ***graph-aware LLMs are also susceptible to feature perturbations, rendering these defenses ineffective***.

One such defense mechanism in GNNs involves making the adjacency matrix trainable, allowing edge weights to adapt during training to mitigate adversarial perturbations. This approach assigns lower weights to edges connecting dissimilar nodes and higher weights to edges linking nodes with greater feature similarity. Several **preprocessing-based** defenses have been proposed building on this approach, such as removing edges between nodes with low feature similarity (Wu et al., 2019) or introducing randomized edge dropping (Dai et al., 2018). However, these defenses rely heavily on clean, unperturbed node features to compute reliable similarity scores. When node features are also perturbed, as in the case of our attacks, such defenses become ineffective and may even exacerbate the problem by connecting incorrect edges. We demonstrate this effect on a random node in the Cora dataset (Figure 8). As observed, while the clean feature similarity graph preserves some original edges alongside introducing some wrong connections, the edges reconstructed by the perturbed features fail to preserve any original connections, leading to entirely incorrect reconstructions.

Another line of work focuses on decoupling reliance on the graph structure. For instance, (Shanthamallu et al., 2021) proposed a defense that trains a surrogate predictor dependent solely on node features and employs an uncertainty matching strategy to extract graph information from the GNN. Similarly, Wu et al. (2021) introduced a co-training framework that trains submodels independently on feature and structural views, allowing knowledge distillation through the inclusion of confident unlabeled data into the training set. However, these approaches also assume access to unperturbed node features, an assumption that does not hold in our setting, where both structure and features are adversarially perturbed. Consequently, structural defense strategies developed for GNNs are not directly applicable to the graph-aware LLM paradigm, which calls for novel defenses tailored to this paradigm.

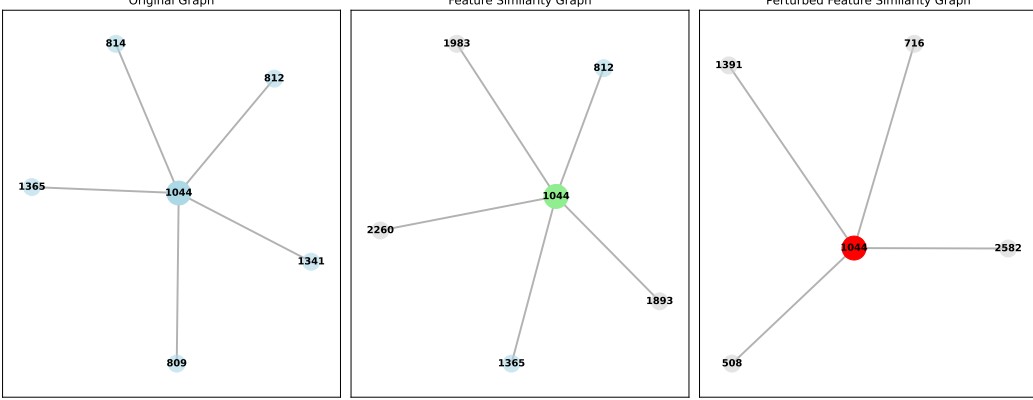

Figure 8: Edges reconstructed for a random node 1044. **Original (left)** is the ground truth edges from the original graph. **Feature Similarity (middle)** shows the edges reconstructed using clean node features, where blue nodes indicate preserved connections from the original graph. **Perturbed Feature Similarity (right)** shows the edges reconstructed using perturbed node features.

