# OpenReview forum: "Adversarial Attacks and Defenses on Graph-aware Large Language Models"
_ICLR.cc/2026/Conference — Submitted to ICLR 2026_

### Official Review · Reviewer_sypf · 2025-10-17

**Soundness:** 3
**Presentation:** 2
**Contribution:** 2
**Rating:** 2
**Confidence:** 5

**Summary:**

This paper investigates the adversarial robustness of graph-aware LLMs, focusing on two representative models: LLAGA and GraphPrompter. The authors adapt existing GNN adversarial attacks to the graph-aware LLM paradigm and discover a novel attack surface specific to LLAGA's node sequence template design. The work introduces imperceptible feature perturbation attacks using homoglyphs and character reordering, demonstrating that graph-aware LLMs are particularly vulnerable to textual feature manipulations. To defend against these attacks, the authors propose GALGUARD, an end-to-end defense framework combining an LLM-based feature corrector with adapted GNN-based structural defenses.

**Strengths:**

1. The paper identifies a previously unknown vulnerability in GraphPrompter and LLaGA.
2. The paper provides a comprehensive analysis of feature perturbation.
3. The propsed GalGuard framework is effective in defending.

**Weaknesses:**

1. My most concern is about the comprehensiveness.
Only two graph-aware LLM methods are evaluated (LLaGA and GraphPrompter, see Section 3), which is insufficient to draw generalizable conclusions about the entire paradigm. The paper claims to investigate "graph-aware LLMs" broadly but provides no evidence for other prominent methods in the taxonomy (Section 2, Appendix A.1). \
The attack for node sequence template seems solely effective for LLaGA, the applicability of it to other Graph-LLM is unclear.

2. The relationship between graph sparsity and vulnerability is observed (denser Cora more vulnerable than sparse PubMed, Section 4.1, page 5) but not rigorously analyzed.
PubMed also differ from Cora and CiteSeer in paper sub-category, text length and many other aspects.
More analysis are required to reach solid conclusions.

3. Missing comparison to existing works exploring Graph adversarial attacks with LLMs. [1-4]

4. The attack is named Mettack. And Table 2 is confusing. Why GalGuard is compared with attack method Mettack?

5. The ArXiv dataset experiment (Appendix A.6, Table 8) shows different attack patterns than smaller datasets, but the paper does not explain why poisoning significantly outperforms evasion on ArXiv (opposite of other datasets). This suggests the findings may not scale consistently.

6. GALGUARD's scalability to graphs with millions of nodes is questionable given its reliance on GPT-4 Turbo API calls for every node's features.


[1] Zhang, Qihai, et al. "TrustGLM: Evaluating the Robustness of GraphLLMs Against Prompt, Text, and Structure Attacks." Proceedings of the 31st ACM SIGKDD Conference on Knowledge Discovery and Data Mining V. 2. 2025.

[2] Guo, Kai, et al. "Learning on graphs with large language models (llms): A deep dive into model robustness." arXiv preprint arXiv:2407.12068 (2024).

[3] Zhang, Zhongjian, et al. "Can large language models improve the adversarial robustness of graph neural networks?." Proceedings of the 31st ACM SIGKDD Conference on Knowledge Discovery and Data Mining V. 1. 2025.

[4] Lei, Runlin, et al. "Intruding with words: Towards understanding graph injection attacks at the text level." Advances in Neural Information Processing Systems 37 (2024): 49214-49251.

[5] Chen, Zhikai, et al. "Exploring the potential of large language models (llms) in learning on graphs." ACM SIGKDD Explorations Newsletter 25.2 (2024): 42-61.

**Questions:**

See weaknesses.

---

### Official Review · Reviewer_h1RW · 2025-10-31

**Soundness:** 3
**Presentation:** 2
**Contribution:** 2
**Rating:** 6
**Confidence:** 3

**Summary:**

The authors are interested in the subject of adversarial robustness in the context of approaching graph-related tasks using Large Language Models (LLMs). The authors start by exploring the application of previously available graph attacks, and then consequently propose a new attack adaptation that exploits putting placeholders into the input node sequence template. The authors also present a novel end-to-end defense strategy, denoted as GalGuard to defend against these perturbations.

**Strengths:**

- As the integration between graph-structured data and LLMs have recently attracted a lot of attention, understanding their adversarial robustness is clearly an important subject.
- This work seems to be (from my understanding and knowledge) the first to tackle this aspect.
- The writing and approach is very clear, specifically the taxonomy makes it easier to understand how to interact with the rest of the attack/defense in this specific context.

**Weaknesses:**

- Some elements in the analysis are not very clear:
    - In Section 4.2, I didn’t understand your claim regarding the difference between LlaGa and GraphPrompter. Specifically, the latter is also based on message-passing framework, and therefore should also be subject to vulnerability when the structure is modified.
- The proposed attack, while novel and clearly clever, is rather adapted for the LLaGa and therefore lacks the desired transferability for other approaches. For instance, if all the graph attack approaches have only focused on changing some specific elements in one specific model, the overall attack literature would still be limited.
- When considering defense methodology, and specifically in the case of node feature-based attacks, you don’t compare or adapt state-of-the-art defense in that specific direction. Namely GCORN [1], which is interested in defending that specific line of attack or AirGNN [2].
- A specific discussion of the complexity of the proposed defense methodology is missing.

—

[1] Bounding the Expected Robustness of Graph Neural Networks Subject to Node Feature Attacks. - ICLR 2024.

[2] Graph Neural Networks with Adaptive Residual. - NeurIPS 2021.

**Questions:**

For the rebuttal, I would suggest that the authors provides and answers the following important elements to further enhance their manuscript:
- Could you provide more clarification and elements regarding the performance difference between LlaGa and GraphPrompter? It would be nice to visualise how does the intermediate output of these elements is affected by the structural perturbations.
- Could you add elements regarding state-of-the-art adversarial defenses (for instance GCORN) that focuses on feature-based attacks?
- Could you provide additional elements regarding the complexity (for instance time complexity) of the proposed GalGuard?

---

### Official Review · Reviewer_GAAK · 2025-10-31

**Soundness:** 2
**Presentation:** 2
**Contribution:** 3
**Rating:** 2
**Confidence:** 3

**Summary:**

This paper claims to be the first systematic study of adversarial attacks and defenses for graph-aware LLMs. It adapts classic GNN attacks (NETTACK, METAATTACK), discovers a new node-sequence injection attack that targets LLAGA’s template, develops imperceptible text feature attacks, and proposes an end-to-end defense (GALGUARD) combining an LLM-based feature corrector with adapted GNN defenses. The experiments use standard benchmarks (Cora, Citeseer, PubMed) to evaluate the robustness of graphLLMs under structural and textual attacks.

**Strengths:**

1. The paper identifies and systematically explores attack surfaces that are unique to LLM-based graph encodings (e.g., node-sequence template injection).
2. The node sequence template injection (NI/SI/MSI) is a convincing new attack tailored to LLAGA’s fixed-shape neighborhood template and shows strong evasion success on small graphs.
3. Proposed a new defensive framework that integrates LLM for feature correction.

**Weaknesses:**

1. The experimental baseline set is narrow: more modern graph attack/defense baselines and a broader set of graph-LLM pipelines (and LLM sizes/types) would strengthen claims, e.g., “Does the Choice of LLM Affect Attack Performance?”
2. The paper states a 10% edge perturbation budget for NETTACK/METAATTACK experiments, but it does not clearly state per-experiment budgets for node-injection attacks (how many nodes can an attacker inject before detection?), or how “unnoticeability” is operationalized for injection and feature attacks.
3. The paper swaps Vicuna and LLaMA-2 in ablations and reports similar trends, but more extensive evaluation over different LLMs (and different GNN encoders / projectors) would strengthen the “choice of LLM has minimal impact” claim. Relatedly, the surrogate model used by NETTACK/METAATTACK is a linearized 2-layer GCN, and authors should justify that this surrogate is adequate to model LLM-projector interactions.

**Questions:**

1. Why is the evasion attack more effective? Please elaborate on the explanations in 4.1 and provide more evidence.
2. How do the surrogate models perform under different attacks??
3. You state that you perturb “only 10% of the edges” for NETTACK/METAATTACK (unnoticeability constraint). For the node-injection attacks (NI/SI/MSI): (a) exactly how many nodes are injected per target / per graph? (b) Is there a budget expressed as a % of total nodes or edges?
4. Is the LLM corrector necessary for GraphLLMs, which uses LLM to predict the results? Please provide an ablation study.  What’s the similarity between the adversarial graph and the sanitized graph in terms of graph structure and features?

---

### Meta-Review · Area_Chair_ub8J · 2026-01-06

**Summary:**

This paper investigates the adversarial robustness of graph-aware LLMs, focusing on two representative models: LLAGA and GraphPrompter. The authors adapt existing GNN adversarial attacks to the graph-aware LLM paradigm and discover a novel attack surface specific to LLAGA's node sequence template design. The work introduces imperceptible feature perturbation attacks using homoglyphs and character reordering, demonstrating that graph-aware LLMs are particularly vulnerable to textual feature manipulations. To defend against these attacks, the authors propose GALGUARD, an end-to-end defense framework combining an LLM-based feature corrector with adapted GNN-based structural defenses. While the core idea is recognized as interesting and the presentation is clear, three reviewers raised substantial concerns that collectively point to a paper that is not yet ready for publication.

**Reviewer Concerns:**

Without a rebuttal, the authors did not provide the necessary clarifications, justifications, or additional experiments requested by the reviewers. Consequently, all major concerns are outstanding and fatal to the paper's acceptance.

**Reviewer Scores:**

Since no rebuttal was provided to address the critiques, it is highly unlikely any reviewer would have raised their scores. The trajectory would likely be neutral or negative.

---

### Decision · Program_Chairs · 2026-01-26

Reject